# OpenReview forum: "BSPA: Exploring Black-box Stealthy Prompt Attacks against Image Generators"
_ICLR.cc/2024/Conference — ICLR 2024 Conference Withdrawn Submission_

### Official Review · Reviewer_vTUC · 2023-10-28

**Soundness:** 3 good
**Presentation:** 3 good
**Contribution:** 4 excellent
**Rating:** 8
**Confidence:** 3

**Summary:**

This paper proposes a stealthy prompt attack which can bypass both the text and image filters. The work successfully raises the ASR by about 5% on four mainstream text-to-generation API.

**Strengths:**

- The attack is meaningful and the experiments are thorough.
- This paper studies a quite practical threat and their methodology is sound to bypass the real-world filters.
- The authors also propose the potential defenses and release the prompt benchmark.

**Weaknesses:**

- The improvement in ASR is not noticeable (about 5%). The authors may provide additional discussion.

**Questions:**

Please see the above weakness part above.

---

### Official Review · Reviewer_aMMq · 2023-10-31

**Soundness:** 2 fair
**Presentation:** 2 fair
**Contribution:** 3 good
**Rating:** 5
**Confidence:** 4

**Summary:**

This paper explores the safety of image generators and develops a black-box stealthy prompt attack (BSPA). The attack aims to generate stealthy prompts that can bypass the detection of image and text filters. The prompts are generated by an LLM model without internal access to the target model. The proposed attack is evaluated on both open-source models and black-box APIs.

**Strengths:**

1. The paper explores malicious prompts for generating NSFW content by text-to-image models, which is a critical topic for generative AI safety.
2. The proposed attack has been validated on real-world online services like Stable Diffusion XL and Midjourney.

**Weaknesses:**

1. The paper is hard to follow. Key details of the proposed attack are missing in the paper. First, how to get the text filter and the image filter? How do we calculate text/image filter scores s_t and s_i? Second, what is the model architecture of the text encoder used in the retriever? How do we get this text encoder? Should this text encoder share a similar architecture or parameters with the image generator? Third, how to use black-box LLM API to generate the modified prompt x_s? Is there any specific prompt used for the generation? Does the LLM API has to be close to the text encoder used in image generator?
2. The novelty of this paper is limited. The attack framework of using LLM and diffusion models to generate NSFW images has been investigated in many recent works (See comment 4). The proposed text retrieval mainly aims to train a surrogate model to simulate the text encoder in the image generator, and then attack the surrogate model to achieve black-box attacks. This idea is also straightforward.
3. The proposed attack leverages black-box LLM APIs to generate texts and images. However, the process can be inhibited if the LLM service provider identifies and blocks the generation of malicious content. If the malicious content is detected frequently, the service provider may stop providing services to the attacker.
4. Recent attacks and defenses on text-to-image generation have not been compared or discussed in the paper. For example, [1-3] have investigated potential attacks for T2I models. Safety filtering [4] and concept removal [5,6] techniques have been proposed to defend against these attacks. The comparison with zeroth-order optimization methods is also missing in the evaluation.
5. The proposed attack does not perform well on black-box APIs. The ASR on SD is merely around 10%. In addition, there is a big discrepancy between filter-based ASR and human-based ASR in the results, which needs further clarification.

[1] Rando, Javier, Daniel Paleka, David Lindner, Lennart Heim, and Florian Tramèr. "Red-teaming the stable diffusion safety filter." arXiv preprint arXiv:2210.04610 (2022).
[2] Qu, Yiting, Xinyue Shen, Xinlei He, Michael Backes, Savvas Zannettou, and Yang Zhang. "Unsafe diffusion: On the generation of unsafe images and hateful memes from text-to-image models." ACM CCS 2023.
[3] Zhi-Yi Chin, Chieh-Ming Jiang, Ching-Chun Huang, Pin-Yu Chen, and Wei-Chen Chiu. Prompting4debugging: Red-teaming text-to-image diffusion models by finding problematic prompts. arXiv preprint arXiv:2309.06135, 2023.
[4] Patrick Schramowski, Manuel Brack, Bj¨orn Deiseroth, and Kristian Kersting. Safe latent diffusion: Mitigating inappropriate degeneration in diffusion models. In Proceedings of the IEEE/CVF Conference on Computer Vision and Pattern Recognition, pp. 22522–22531, 2023.
[5] Kumari, Nupur, Bingliang Zhang, Sheng-Yu Wang, Eli Shechtman, Richard Zhang, and Jun-Yan Zhu. "Ablating concepts in text-to-image diffusion models." In Proceedings of the IEEE/CVF International Conference on Computer Vision, pp. 22691-22702. 2023.
[6] Gandikota, Rohit, Joanna Materzynska, Jaden Fiotto-Kaufman, and David Bau. "Erasing concepts from diffusion models." ICCV. 2023.

**Questions:**

Please describe the details of the proposed attacks as mentioned in Weakness 1.

---

### Official Review · Reviewer_XfJe · 2023-11-02

**Soundness:** 3 good
**Presentation:** 2 fair
**Contribution:** 3 good
**Rating:** 5
**Confidence:** 3

**Summary:**

This work proposes an automatic tools for generating stealthy prompt attacks against the state-of-art text-to-image generators, BSPA. This approach enables the automatic generating stealthy prompts against image generators using Large Language Models (LLMs). This approach is simple but effective and efficient for generating toxic images through zero-order gradient estimation. The results show this approach is effective across various SOTA generative models.

**Strengths:**

1. This approach is practical and effective.
2. The motivation is clear and reasonable.
3. The evaluation is comprehensive and the results are effective.

**Weaknesses:**

1. My biggest concern is that I think the effectiveness of this approach highly relies on the safety of used Large Language Models (LLMs). Just as an example, ChatGPT-4 has some defense to prevent generating toxic sentences or sentences describing sexual or unhealthy scene. Therefore, how can you leverage existing SOTA LLMs for crafting stealthy prompt.

2. Missing a lot of details.



If you can answer my questions and address my concerns, I will raise my score.

**Questions:**

1. What LLM models do you use ?

2. How do you set the Harmful loss function, i.e., $L_{harm}$ ?

3. How do you train the text/image filters?

4. How do you initialize the attack sentence?

5. How do you define and evlauate the stealthness of your approach. From my perspective, i don't think the prompt is stealthy at all.

---

### Official Review · Reviewer_oGes · 2023-11-02

**Soundness:** 2 fair
**Presentation:** 2 fair
**Contribution:** 2 fair
**Rating:** 3
**Confidence:** 5

**Summary:**

This paper proposes a novel approach to improving the safety of image generators by crafting stealthy prompts tailored for them. The proposed black-box stealthy prompt attack (BSPA) requires no internal access to the model's features, making it applicable to a wide range of image generators. The authors introduce a text retriever that provides a larger retrieval space for retrieving sensitive words and design an optimization scheme, including pseudo-labeling and loss function. They also construct a benchmark to evaluate the model’s ability to reject NSFW content.

**Strengths:**

+ originality: beyond the human instruction attack based on large language model, this paper proposes to distills the knowledge of an image and text toxicity filter into a retriever, whose output guides large language model to generate stealthy and malicious prompt.

+ significance: The whole pipeline is model-agnostic and it approximates the tactics of the attacker against commercial APIs, which inspires the AI safety community to design a universal defense strategy for the overall black-box attacks.

**Weaknesses:**

- the rationality of retriever: 1). this paper claims that the retriever generates diverse sensitive words, while in the implementation, it actually samples words from 50 candidates, which reside in a very limited search space. 2). the approximation of the attacker is coarse, since the retriever only returns top-3 words from 50 candidates for the following inference of large language model, while in the realistic scenarios, the attacker could apply prompt dilution [1], misspelled or malformed inputs, or any other strategies to evade the safety filter.

- the upper bound of the whole pipeline depends on the knowledge of the selected open-source image and text filter. Since the image or textual filters of the commercial APIs such as DALLE-3 are not accessible, the transferability of the outputs obtained by their pipeline remains a question.

- the fairness of evaluation metric: this paper includes the proportion of their generated images rejected by the image filter as the part of attack success rate (ASR). However, in the real case, commercial APIs would not return those images if their image filter recognizes its NSFW content. So, including such metric in ASR may by not fair.

- the experiment results show no great advantage over human instruction attack, TLA. in Table 1, the toxic rate of TLA is lower than BSPA, which means the output of BSPA is less stealthy than TLA. As for ASR_hum, BSPA is 3.8\%, gaining little advantage than 1.66\% of TLA. Moreover, in Table 3, with their proposed RSF filter, the explicit attack achieves 3.6\% better performance than BSPA, which raises the doubt of the generality of the RSF filter.

[1] Javier Rando, Daniel Paleka, David Lindner, Lennard Heim, and Florian Trame`r. Red-teaming the stable diffusion safety filter. In NeurIPS ML Safety Workshop, 2022.

**Questions:**

- the analyses of training dynamics of the retriever: any experiment could help here if it shows during the training process, the receiver actually learns to adopt some strategy to evade the safety filter.

- missing details of evaluation of black box API. in Table 3, what is the config of the used image filter when calculating ASR_fil of MJ-demo?

- missing experiments about more advanced commercial APIs, such as DALLE 3.